# The Preventive Role of Regular Physical Training in Ventricular Remodeling, Serum Cardiac Markers, and Exercise Performance Changes in Breast Cancer in Women Undergoing Trastuzumab Therapy—An REH-HER Study

**DOI:** 10.3390/jcm9051379

**Published:** 2020-05-07

**Authors:** Katarzyna Hojan, Danuta Procyk, Dorota Horyńska-Kęstowicz, Ewa Leporowska, Maria Litwiniuk

**Affiliations:** 1Department of Rehabilitation, Greater Poland Cancer Centre, 61-866 Poznan, Poland; hkdorota@gmail.com; 2Central Labolatory, Greater Poland Cancer Centre, 61-866 Poznan, Poland; danuta.procyk@wco.pl (D.P.); ewa.leporowska@wco.pl (E.L.); 3Department of Chemotherapy, Greater Poland Cancer Centre, 61-866 Poznan, Poland; maria.litwiniuk@wco.pl; 4Department of Oncologic Pathology and Prophylaxis, Poznan University of Medical Sciences, 61-866 Poznan, Poland

**Keywords:** rehabilitation, prevention, heart failure, oncology, cardiotoxicity

## Abstract

Cardiotoxicity is known as a severe clinical problem in oncological practice that reduces the options for cancer therapy. Physical exercise is recognized as a well-established protective measure for many heart and cancer diseases. In our study, we hypothesized that supervised and moderate-intensity exercise training would prevent heart failure and its consequences induced by trastuzumab therapy. The aim of this study was to examine the effect of physical training on ventricular remodeling, serum cardiac markers, and exercise performance in women with human epidermal growth receptor 2 (HER2+) breast cancer (BC) undergoing trastuzumab therapy. This was a prospective, randomized, clinical controlled trial. Forty-six BC women were randomized into either an intervention group (IG) or a control group (CG). An exercise program (IG) was performed after 3–6 months of trastuzumab therapy at 5 d/week (to 80% maximum heart rate (HRmax)) for 9 weeks. We then evaluated their cardiac function using echocardiography, a 6-Minute Walk Test (6MWT), and plasma parameters (C-reactive protein (CRP), myoglobin (MYO), interleukin-6 (IL-6), alanine aminotransferase (ALT), aspartate aminotransferase (AST), and creatine kinase (CK)). After the physical training program, we did not observe any significant changes in the left ventricular (LV) ejection fraction (LVEF) and 6MWT (*p* > 0.05) in the IG compared to the CG (decrease *p* < 0.05). The differences in the blood parameters were not significant (*p* < 0.05). To conclude, moderate-intensity exercise training prevented a decrease in the LVEF and physical capacity during trastuzumab therapy in HER2+ BC. Further research is needed to validate our results.

## 1. Introduction

Breast cancer is the most common cancer affecting women, causing 24.6% of female cancers and resulting in 15% of all female cancer deaths worldwide [1]. The last two decades of cancer research showed that early screening and advances in onco-therapy management have resulted in noticeable progress, as evidenced by an increase in the survival rates among breast cancer (BC) patients [1]. Despite promising preclinical models for different strategies for BC therapy [2,3,4,5,6], monoclonal antibodies against human epidermal growth receptor 2 (HER2) expression, such as trastuzumab, are currently clinically indicated to ameliorate survival in this group of patients [7,8]. However, despite the favorable perspective of this kind of BC treatment, various side effects of this therapy can be observed [9].

One of the negative consequences is cardiotoxicity, which causes a considerable decrease in physical performance, limits anticancer therapy effects, and may also contribute to an increase of mortality in this group of patients [10,11]. About one third of BC women have a positive expression of HER2 proteins [1], which increase the probability of tumor cell invasion or survival of metastatic cancer. On the other hand, there are cases of women who evolved left ventricular (LV) dysfunction (in 4%–18% patients) [3,12] or heart failure (2%–4% of treatment patients) [3] following trastuzumab treatment. The current recommendations indicate using a measurement of the LV ejection fraction (LVEF) to evaluate cardiac function in patients who have been administered trastuzumab therapy [13].

Cardiotoxicity occurs irrespective of medicine dose, is not associated with structural modification in cardiomyocytes, and is completely convertible after discontinuation of this treatment [12,14,15]. LVEF has considerable limitations as well as a low sensitivity to reveal an early stage of cardiotoxicity [13]. To better monitor cardiac function, a multifactorial approach is recommended. This approach should use echocardiographic measurements, cardiac injury parameters, and clinical tests to measure cardiorespiratory tolerance [13,15,16,17,18,19]. Currently, the implementation of preventive solutions to optimize heart care during potentially cardiotoxic therapy in cancer patients is limited [17,18,19]. The knowledge of physical exercise activity guidelines in the population of breast cancer patients also remains limited [20,21]. Physical exercise is acknowledged as a safe, feasible, and nonpharmacological practice for the prevention of many cardiovascular risk factors [13,14], as well as an important supportive measure during cancer care [22,23]. The available data underline the positive influence of regular physical exercise on physiological and psychological results during BC treatment [24,25] and emphasize its significant role in post-cancer recovery [26,27]. Moreover, regular physical activity might limit the cardiotoxicity associated with oncological treatment. However, clinicians who promote physical activity during cancer treatment (according to many recommendations) should consider cardiac function, capacity or fatigue, and other comorbidities that restrict physical performance [28].

In our study, we hypothesized that supervised exercise training would prevent heart failure resulting from trastuzumab therapy. This would happen by maintaining or increasing the LVEF and improving the myocardial parameters and physical exercise performance, which, in turn, would improve the inflammatory and metabolic parameters. 

Therefore, the aim of this feasibility study was to examine the effect of supervised endurance and strength training on ventricular remodeling, selected plasma markers, and exercise tolerance in HER2-positive BC women undergoing trastuzumab therapy.

## 2. Materials and Methods

### 2.1. Study Design

This was a two-arm parallel prospective, clinical, randomized, controlled trial. The Bioethics Committee at Poznan University of Medical Sciences (No. 785/14) approved the study protocol. All patients participating in the study had to obtain approval from an oncologist and cardiologist. Additionally, a research physiatrist provided a detailed explanation of the study’s physical exercise program to the patients and then obtained their voluntary written informed consent to participate in this trial.

### 2.2. Setting and Participants

The study was performed at the outpatient rehabilitation ward at the provincial cancer hospital from August 2017 to December 2019. A screening for BC with the HER2-positive receptor performed at the hospital was used to identify prospective study participants. Sixty-eight female BC patients undergoing trastuzumab treatment were originally selected according to the study criteria. Trastuzumab was administered at 8 mg/kg i.v. over 90 min (loading dose) and then at 6 mg/kg i.v. every 3 weeks (i.v. infusion over 30–90 min) over one year of treatment. The inclusion criteria were as follows: BC women with an HER2-positive receptor confirmed histologically after chemo- and radiotherapy, aged 18–75 years, in good general health according Eastern Cooperative Oncology Group (ECOG) performance status 0–2, with good heart function and LVEF (above 60%), and normal function of the liver (normal levels of aspartate aminotransferase (AST) and alanine aminotransferase (ALT)) and the kidneys (with creatinine clearance above 60 mL per minute).

To ensure that the participants formed a homogenous group, we only allowed patients between 3–6 months of trastuzumab treatment in the trial. We excluded women from the study if they had HER2 negative BC, distant metastases, disease progression that necessitated the introduction of radiation treatment or chemotherapy, cardiac diseases resulting in circulation failure (above the second class of the New York Heart Association (NYHA)), autoimmune disease (e.g., systemic lupus erythematosus or rheumatoid arthritis), lung disease (chronic obstructive pulmonary disease), or uncontrolled asthma. We eliminated from the study those patients who had progression in heart failure causing a significant decrease in LVEF (below 10% during the observation or under 55% of the total value), and resting oxygen saturation (SaO_2_) ≤ 92%. Another contraindication was malnutrition (a body mass index below 18 kg/m^2^) or weight loss of >10% during the previous 3 months, psychiatric or cognitive disorders making it impossible for the patient to comply with regular study recommendations, or pregnancy (or breastfeeding) and death. Furthermore, a given participant was not included if she was being treated with an angiotensin receptor blocker, an angiotensin converting-enzyme inhibitor (ACE), or a beta-blocker. The study protocol was described in accordance with the Consolidated Standards of Reporting Trials (CONSORT) statement (Figure 1).

### 2.3. Randomization and Blinding Procedures

A computer-generated list of random numbers was used to allocate the study patients to one of the study groups, the intervention group (IG) or the control group (CG), according to simple randomization procedures. Sequentially numbered opaque envelopes were used for concealed randomization. These envelopes contained group assignments and were provided to the participants after the baseline assessment. All data of the participants who did not complete the study protocol were excluded from the statistical analysis. Our research was not fully blinded. However, both the patients and the physiatrist were blinded to the group allocation before the first assessment. The physiatrist collected the self-reports, conducted physical-capacity tests, and explained the exercise program to the participants in IG. The cardiologist, laboratory assistants, and study statistician remained blinded at all times.

### 2.4. Measurement Scheme

The BC participants were also assessed in terms of changes resulting from the trastuzumab therapy. These assessments were performed during the same time period as those pertaining to heart failure and physical fitness. In both study groups, the following schedule was used: Baseline (Assessment Time 1—T1) before the start of the study intervention involving regular exercise after a minimum of 3 months of trastuzumab therapy onset and Assessment Time 2 (T2) after 9 weeks of physical exercise training or usual care.

### 2.5. Physical Exercise Program

Physical training in IG started after a minimum of 3 months but no later than 6 months after trastuzumab treatment. The program was based on endurance and strength exercises and aimed at assessing the impact of physical exercise on the negative results of oncological treatment as observed before the initiation of training. This program took place at the outpatient rehabilitation ward. The physical activity was moderate, with a maximum heart rate (HRmax) of 80% (according to the calculation HRmax = 220–age) during the exercises [29]. All training sessions in the IG consisted of five exercise sessions per week for 9 weeks.

Endurance training involved choosing a maximum of two forms of physical activity in a one-day session from several options, such as brisk walking, running on a treadmill, and various cycling activities. This part of the daily training program lasted roughly 45–50 min according to the following scheme: 2 min of warm-up, 45 min of one or two aerobic activities, and a 3 min relaxation period.

Additionally, women in the IG performed strength exercises to increase their muscle mass and strength. This part of the daily session (5 days per week) lasted approximately 40–45 min. The resistance exercise sessions based on isometric, concentric, and eccentric training consisted of one to three sets of 8–10 repetitions of selected exercises in different positions for the trunk, upper body, and leg muscles. To ensure the progressive development of strength, we used different tools: elastic resistance bands (1 kg, 2 kg, 2.5 kg, 3.5 kg, and 4.5 kg per 100% extension), a TRX gym system, an indoor rower, adjustable dumbbells, and medicine balls. The weight training program intensity was progressively raised by increasing the amplitude of the movement, introducing more strenuous exercises, or shifting the velocity of the concentric performance [29].

Women in the IG were instructed not to participate in any other forms of weight exercise during the study period. Their compliance was verified via a daily interview with a physiotherapist who conducted the training. The exercise program was implemented in groups (exercises on treadmills or cycle ergometers, supervised by a therapist) and individually (strength training with the assistance of a physiotherapist). No more than two days were recommended as the maximum break from training during the study (9 weeks). Participation in the exercise program in the IG was verified through a rehabilitation card that was checked by a research physiatrist after the end of the study.

### 2.6. Usual Care

BC women in the CG performed the usual daily activities indicated for heathy people [30]. The physiatrist and physiotherapist provided the CG participants with physical activity instructions via printed materials. The women were asked not to begin any formal exercise training and to only perform their daily activities at home for no more than 150 min per week. The results obtained in this group were analyzed based on the Godin–Shephard Leisure-Time Physical Activity Questionnaire [31].

### 2.7. Outcome measures

The primary outcomes were the differences in cardiac function measured with a capacity test over the nine weeks of the exercise program. The secondary outcomes were changes in the blood biomarkers specific for heart disease.

### 2.8. Measurements

#### 2.8.1. Anthropometric Measurements

Anthropometric parameters such as weight and height were used to determine each participant’s body mass index (BMI). During the measurements, the participants were barefoot and wore only underwear. Body mass was estimated using a weight calibrated to the nearest 0.1 kg. The height was marked using a calibrated length board to the nearest 0.01 m. The BMI was calculated as the body mass divided by the height in meters squared (kg/m^2^).

#### 2.8.2. Measurements of Cardiac Function

Cardiac function was measured using Doppler echocardiography (Philips EPIQ7, Philips Healthcare, Andover, MA, USA) in accordance with the recommendations by the European Association of Echocardiography and the American Society of Echocardiography [32]. The assessment was based on the mean of three representative beats. During the echocardiographic examination, we performed two-dimensional, M-mode, and Doppler flow measurements. The same echocardiographer, who had knowledge of the various abnormalities, performed all echocardiographic imaging tests and was blinded to all data. The LVEF was assessed using 2- and 4-chamber views via Simpson’s biplane rule, and the left atrial volume was indexed by the area of the body surface. The global longitudinal strain was measured by speckle-tracking 2-dimensional echocardiography. These values were shown as positive to allow for an easier interpretation of the data, despite the fact that negative values are considered normal. Another parameter was the right ventricular function, which was assessed using the tricuspid annular plane systolic excursion. Additionally, we observed the blood pressure and heart ratio. Doppler-derived LV diastolic inflow was performed to measure the E and A peak velocities and their ratios, E/A.

#### 2.8.3. Physical Capacity Tests

Physical performance was evaluated with a 6 min walk test (6MWT), which is commonly used among cancer patients for the assessment of aerobic capacity [33]. We carried out the 6MWT according to the guidelines of the American Thoracic Society [34] in a hospital corridor after a reset (cooldown) pause. We measured the 6 min walk distance (6MWD), metabolic equivalents of tasks (METs), and dyspnea after a test using a modified Borg scale [34]. As evidence suggests that 3.5 mL/kg/min is not an accurate representation of the resting metabolic rate in the general population, we measured the metabolic rate as METs = VO_2_/pretest. The pretest metabolic rate was calculated as the mean VO_2_ in the 1–2 min prior to 6MWT initiation. For this assessment, we used medical gas (Oxycon Mobile, CareFusion, Germany).

#### 2.8.4. Measurements of Plasma Parameters

The blood was drawn by a venous puncture between 7 a.m. and 9 a.m. after overnight fasting. Centrifugation was used to separate the plasma (15 min at 3500 rpm), and the samples were stored at −80 °C until analysis. We measured the myoglobin (MYO), interleukin-6 (IL-6), alanine aminotransferase (ALT), aspartate aminotransferase (AST), creatine kinase (CK), and isoenzyme myocardial band (CK-MB) as important parameters for heart muscles and heart failure, as these measures correlate with physical activity. These parameters were determined quantitatively in the patients’ blood serum using a COBAS 6000 analyzer (Roche Diagnistics International Ltd, Rotkreuz, Switzerland). The AST, CK, and CK-MB tests were performed using kinetic methods with absorbance measurements. High-sensitivity C-reactive protein (hsCRP) was determined using an immunoturbidimetric method enhanced with latex particles. The IL-6 and MYO tests were performed via an electrochemiluminescence method (ECLIA) using mouse antibodies labeled with biotin and ruthenium and coated with streptavidin (specific for IL-6 and MYO, respectively).

### 2.9. Statistical Analysis

The general statistical values, including the mean, standard deviation (SD), median, and interquartile range (IQR), were used for continuous variables. Comparisons of the quantitative parameters in the two groups were conducted with a Mann–Whitney test. Analysis of the two repeated measures was conducted with a paired Wilcoxon test. Spearman’s correlation coefficient was used to assess the correlations between the quantitative variables. Analyses were conducted at a 0.05 level of significance. R software, version 3.6.1 (R Foundation for Statistical Computing, Vienna, Austria, 2019) was used [35].

## 3. Results

### 3.1. Study Group

Figure 1 presents the flow chart of the study protocol. The study program was completed by 47 BC patients receiving trastuzumab therapy. The other participants who were originally qualified discontinued the study due to one of the following reasons: resignation from the study, disease progression, lack of adherence to the exercise program prescribed, or death. We analyzed the data from 47 consecutive patients with HER2-positive BC. Adherence to the exercise training was 98.7% in the IG. Four participants had a two-day break, and seven women had only a one-day break during the study program.

Table 1 shows the baseline characteristics of the study participants at the first cardio-oncology visit and during the baseline rehabilitation assessment. The BC participants whose mean age was 54.5 ± 6.05 years (standard deviation) were in the early stages of the disease. The immunohistochemistry test results indicated that all patients had HER2-positive disease. Thirty-four BC participants (72.3%) were hormone-positive, and 26 (55.3%) had cancer in the left breast. Twenty-six women (55.3%) had been treated with radiotherapy. Anthracycline therapy was previously administered to 38 patients (80.8%) according to the standard protocols for HER2-positive BC. Most of them received four cycles of AC chemotherapy (i.e., doxorubicyn 60mg/m^2^ and cyclophosphamide 600 mg/m^2^) every 3 weeks and 12 cycles of weekly paclitaxel (80 mg/m^2^) plus trastuzumab every 3 weeks, followed by trastuzumab alone, to complete a year of treatment. Patients with small tumors and negative nodes (19.2% of all BC) received weekly paclitaxel plus trastuzumab every 3 weeks for 12 weeks, followed by trastuzumab every 3 weeks for 9 months.

The patients presented histories of the following conditions: diabetes in 1 patient (2.1%), dyslipidemia in 6 patients (12.7%), hypertension in 8 patients (17%), and smoking in 8 patients (i.e., 17% were former smokers). One patient (2.1%) had previous heart failure; however, a cardiologist enrolled her into the trastuzumab treatment. Clinically significant heart failure did not occur in any of the study participants. We identified the first class of New York Heart Association (NYHA) in 42 women (89.3%), and the second class in only two participants (4.2%). Table 1 presents the characteristics of the study patients in the IG and CG.

### 3.2. Analysis of the Echocardiography Results

The analysis of the echocardiography results and comparison of the cardiac function in both study groups are presented in Table 2. There was a statistically significant decrease of the LVEF (*p* < 0.05) in the CG compared to the IG, which did not present any significant changes. The other echocardiography parameters did not change significantly.

### 3.3. Results of the Physical Capacity Test

Table 3 shows the results of the 6MWT in the study groups. The 6MWD decreased significantly (*p* < 0.05) in the CG after study observation. The other parameters did not change significantly (*p* > 0.05) in either group, but the trends of the changes observed were opposite between the IG and CG.

### 3.4. Analysis of the Blood Parameter Results

The comparison of blood test results between the study groups is presented in Table 4. There were no statistically significant differences between the study groups for the measurements of blood parameters.

### 3.5. Correlations between Study Parameters

Table 5 presents the correlations between the study parameters (all blood biomarkers with BMI, 6MWT, and echocardiography parameters) before and after the study in the two study groups. We presented only the correlations that were statistically significant during the study period. In the IG, the CK-MB correlated positively (r > 0) and significantly (*p* < 0.05) with the BMI and Borg scale but negatively (r < 0) and significantly (*p* < 0.05) with the 6MWD, LVEF, and MET at the beginning of the study. After physical activity intervention (at the second assessment), the CK-MB correlated significantly (*p* < 0.05) only with the BMI (r > 0). In the CG in Assessment I, only IL-6 correlated significantly and negatively (*p* < 0.05; r < 0) with the 6MWD and MET. After the study, these correlations were not statistically significant, but the CK-MB correlated negatively and significantly (*p* < 0.05; r < 0) with the 6MWD and MET.

## 4. Discussion

To prevent BC recurrence, it is essential to estimate the potential of new approaches for supportive care. One potential strategy is physical exercise training. The American Cancer Society and American College of Sports Medicine recommended the implementation of exercise training during the entire duration of oncological treatment [22,29]. Additionally, physical training is recommended as a prevention and treatment for many cardiac illnesses [36,37].

Our feasibility study was designed to answer questions regarding the application of new preventive procedures related to everyday life activities to help maintain good physical performance and cardiac health during cancer treatment in BC patients. The overexpression of the HER2 protein occurs in approximately one third of BC patients [11,12,13]. It is not well known whether HER2-positive BC women who have received chemotherapy with trastuzumab can safely perform physical activity with good tolerance to prevent heart failure symptoms. There is little information and a lack of guidelines on preventative care and exercise available to BC patients with the HER2-positive receptor during trastuzumab treatment.

Our study is one of the first prospective, clinical studies for this group of patients that implemented regular exercise training for the prevention of cardiotoxicity in the early stages of this treatment (between 3–6 months). Our exercise program was based on the American College of Sports Medicine recommendations [22,29]. To conduct the heart function evaluation, we used transthoracic Doppler echocardiography according to the relevant guidelines and standards [32], which was performed only by one physician (a cardiologist) to avoid measuring errors. After the evaluation of the patients, we observed a statistically significant change only in one echocardiography parameter (a decrease in the LVEF in the CG).

This parameter (LVEF) was recommended as a standard clinical test for detecting cardiotoxicity during oncological treatments [17]. However, our results did not reach the level that denotes cardiotoxicity according to the American Society of Echocardiography and the European Association of Cardiovascular Imaging, i.e., a significant decrease in the LVEF >10% or an LVEF below 53% [16,17]. The participants who engaged in regular training (IG) did not present any significant changes in their heart echocardiography results.

During the study, we did not notice any significant changes in the left atrial volume. This likely stemmed from the fact that the small degree of LV dysfunction (only in CG) did not have a sufficient influence on the LV filling pressure to cause dilatation in the left atrium. Another echocardiography parameter, tricuspid annular plane systolic excursion, which was used as a marker of right ventricular systolic function, also did not change during our observations.

Previously, only Haykowsky et al. [38] demonstrated a positive impact from regular physical exercise on the selected cardiovascular parameters in breast cancer patients undergoing chemotherapy with trastuzumab. Following the intervention, there was a significant LVEF decrease and a significant increase in the end diastolic and end systolic volumes. The authors [38] used magnetic resonance imaging (MRI) to measure the cardiac function during 4 months of trastuzumab treatment combined with three days per week of physical exercise (0.5–1 h per session), with a VO_2_max of approximately 60%–90%. At the end of the study, the researchers concluded that the intensity of their proposed physical exercise program was not suitable to prevent LV remodeling [38].

Our study was shorter (only nine weeks) and involved a combination of mixed-type training (aerobic and resistance training) for 90 min a day for 5 days per week with supervision by a physiotherapist. We performed our clinical observations in a rehabilitation ward in a cancer hospital, which translated into 98.7% adherence among the IG patients and most likely influenced the reliability of our results. In the conclusion section of their study, Haykowsky et al. [38] suggested that the poor outcomes in terms of cardio-protection could have been due to the patients’ low adherence to the exercise regime (59%).

To measure the functional capacity, we used the 6MWT, which is a well-tolerated, practical, and useful tool with worldwide recommendation for the cardiorespiratory domain [33,39]. Many studies have demonstrated that the 6MWT and aerobic fitness are predictors for morbidity or mortality in patients with heart failure [39]. After the end of the study, we observed a decrease in the 6MWD in the CG as opposed to the IG (no statistically significant changes). Other results of the 6MWT did not change significantly. Waart et al. [40] suggested that a moderate to high intensity exercise program can limit decreases in the physical fitness, muscle strength, and symptom burden, whereas low intensity training may be preferable for patients undergoing chemotherapy.

Our results yielded clinically important observations that should be investigated in further research. The secondary outcomes of our study were changes in the blood biomarkers specific for heart disease [41]. We used the MYO levels as an indicator of myocardial damage. This is a specific, contractile protein that has high sensitivity to myocardial injury and necrosis [42,43,44]. Our results did not show any significant differences in this parameter.

Another measured parameter, whose elevated circulating levels are connected with congestive heart failure, was IL-6. This parameter may be involved in the damage of the myocardium or cause other symptoms of heart failure. For example, a high level of circulating IL-6 is a strong predictor of clinical outcomes associated with advanced LV dysfunction, which causes heart failure in myocarditis, cardiomyopathy, or allograft rejection [45,46]. Weakened cardiac function is associated with an increased production of IL-6 [45]. Elevated levels of IL-6 affect the late phase preconditioning that provides cardio-protection [47].

To exclude acute inflammation, we measured the hsCRP levels and did not notice any practical or significant changes in the study groups during the observations. Some statistically insignificant differences between the study groups (IG vs. CG) presented opposite trends for change. Many authors have reported an increase in the serum concentration of anti-inflammatory cytokines after different types of exercise [48,49,50,51], which is connected with exercise intensity and indirectly represents the muscle mass involved in the contractile activity [49,51]. Our results showed an increase (but not statistically significant) in the levels of IL-6 in the IG after exercise intervention, which was not observed in CG.

Similar results were obtained in previous trials [48,49,50,51]. The other blood parameters we measured were CK and its isoenzyme CK-MB, which are widely used markers in the diagnosis of cardiomyopathies [41]. CK and CK-MB are reliable markers of myocardial necrosis and have a high sensitivity to detect infarct extensions. On the other hand, physical training can influence the manifestation of CK in blood serum [52]. The CK activity measured during training was also different before and after and changed due to different protocols, intensities, and levels of training [53]. Our results showed that the CK levels in the IG increased after the end of the study but that this increase was not statistically significant. On the other hand, the CK concentration decreased in the CG. Chevion et al. [54] observed that, unlike individuals who did not exercise regularly, the subjects who performed regular, intensive exercise training had significantly elevated base levels of CK. Brancaccio et al. [53] suggested that intensive physical exercise can have an impact on skeletal and heart damage markers (e.g., CK-MB activity), particularly when the training is long or strenuous. Our results did not reveal significant changes. However, the trends displayed by the IG and CG after physical training were similar to the observations made by other researchers [53,54,55].

The other blood markers we measured were AST and ALT. Elevation of AST and ALT is usually connected with liver and skeletal muscle injuries, as well as heart attacks [56]. Other authors suggested that increased levels of AST and ALT in professional athletes do not result from liver abnormalities but rather from the activity of muscle cells that release those enzymes [57,58]. Our study showed that the AST and ALT did not change significantly in either group, despite the mixed-type physical training that was used.

We noticed some significant correlations in both groups. The CK-MB had a negative correlation with the 6MWD and MET units, which was clinically useful for our patients.

### Limitations and Strengths

One of the limitations in our study was the small number of study participants, which makes the work a feasibility study. More data on medical scans (e.g., for body composition) is desirable. Second, the study observation time was nine weeks; thus, further observations are required. However, this study has some strengths. First, our study was a clinical, prospective, randomized controlled trial. Second, our technical staff were blinded to the group allocation of the participants. Finally, the exercise protocol was standardized by strict parameters for the volume of training sessions (heart rate, repetitions, sets, etc.).

## 5. Conclusions

Regular, moderate intensity, supervised exercise training may constitute a primary prevention treatment for heart failure in HER2-positive BC women who might be at risk of cardiotoxicity during the early period of trastuzumab therapy. To ensure the success of this program, it is essential to facilitate adherence and motivation among the participants. Future research is needed to test our interventions and results more precisely and to determine what practical guidelines should be given to this group of cancer patients.

## Figures and Tables

**Figure 1 jcm-09-01379-f001:**
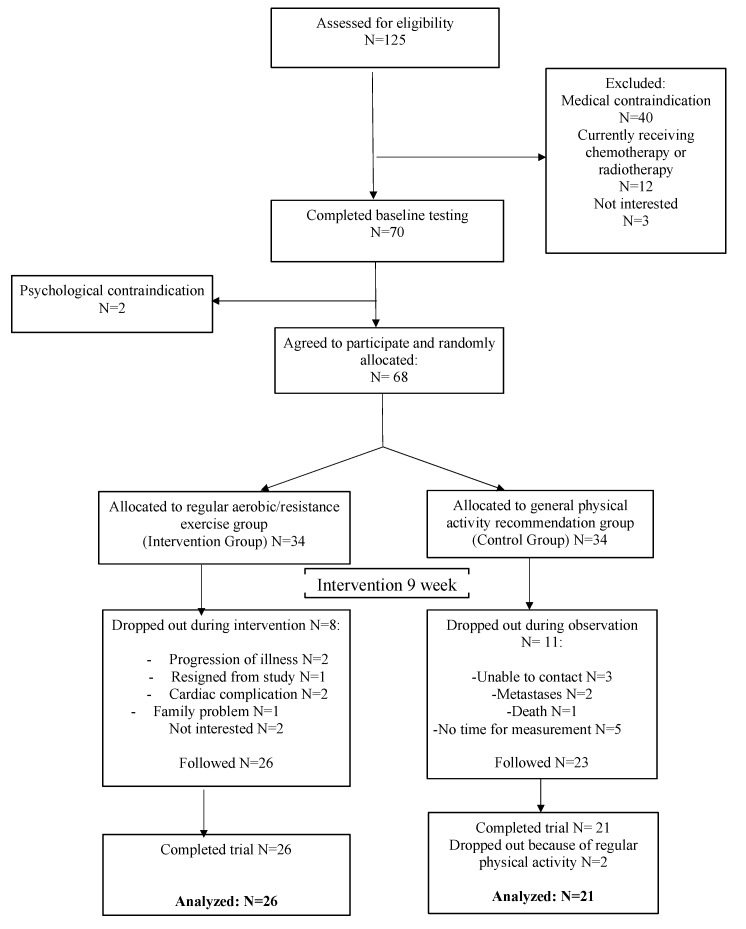
CONSORT diagram.

**Table 1 jcm-09-01379-t001:** Study groups.

Parameter	Intervention Group (IG) *n* = 26	Control Group (CG) *n* = 21	*P_IG vs. CG_*
**Anthropometrical data (Mean ± SD)**			
Age (years)	54.44 ± 6.29	54.64 ± 5.26	0.961
Weight (kg)	65.69 ± 5.25	68.73 ± 2.53	0.182
Height (m)	1.65 ± 0.04	1.68 ± 0.04	0.937
BMI (kg/m^2^)	24.35 ± 2.8	25.35 ± 1.89	0.429
**Medical history (*n* (%))**			
Stage of cancer			
IB	2 (7.7%)	0	-
IIA	9 (34.6%)	10 (47.6%)	-
IIB	12 (46.1%)	11 (52.4%)	-
IIIA	3 (11.5%)	0	-
**Surgical treatment**			
Side of operated breast - Left	15 (57.7%)	11 (52.4%)	-
Side of operated breast - Right	11 (42.3%)	10 (47.6%)	--
BCT	13 (50%)	14 (66.6)	
Mastectomy	5 (19.2%)	2 (9.5%)	-
Mastectomy with reconstruction	8 (30.7%)	5 (23.8%)	-
**Additional oncological treatment**			
Previous anthracycline treatment	20 (76.9%)	18 (85.7%)	-
Previous radiotherapy	16 (61.5%)	13 (61.9%)	-
Hormonal therapy	18 (69.2%)	16 (76.2%)	-
**Other comorbidities**			
Diabetes	1 (3.8%)	0	-
Dyslipidemia	4 (15.4%)	2 (9.5%)	-
Hypertension	5 (19.2%)	3 (14.3%)	-
Previous heart failure	1 (3.8%)	0	-
**Additional data**			
History of smoking	6 (23%)	2 (9.5%)	-
NYHA functional class - I	22 (84.6%)	20 (95.2%)	-
NYHA functional class - II	4 (15.4%)	1 (4.7%)	-

P—Mann–Whitney test; BCT—Breast Conserving Therapy; NYHA—New York Heart Association functional classification; BMI—body mass index.

**Table 2 jcm-09-01379-t002:** Comparison of the cardiac function assessment in both study groups.

Parameter	Time	Intervention Group (IG)		Control Group (CG)		*P_IG vs. UG_*
		Mean ± SD	*P_T1vsT2_*	Mean ± SD	*P_T1vsT2_*	
LVEF (%)	T1	65.69 ± 5.02	0.143	63.9 ± 2.72	0.009 *	0.126
	T2	64.88 ± 5.81		59.82 ± 4.02		0.003 *
GLS (%)	T1	17.5 ± 2.5	0.946	17.3 ± 2.5	0.782	0.746
	T2	17.6 ± 2.5		16.8 ± 2.5		0.422
LAVI (mL/m)	T1	24.6 ± 2.5	0.244	23.8 ± 2.5	0.864	0.624
	T2	26.2 ± 2.5		24.2 ± 2.5		0.484
RVEF (%)	T1	53.3 ± 6.5	0.488	52.8 ± 7.5	0.788	0.812
	T2	54.2 ± 5.2		52.1 ± 6.6		0.575
TAPSE (mm)	T1	20.3 ± 2.5	0.764	21 ± 3.1	0.824	0.686
	T2	21.2 ± 2.2		20.1 ± 3.6		0.755
E/A	T1	1.4 ± 0.5	0.75	1.4 ± 0.6	0.711	0.944
	T2	1.5 ± 0.5		1.5 ± 0.4		0.998
HR (bpm)	T1	65.3 ± 5.5	0.675	65.3 ± 6.4	0.712	0.997
	T2	64.6 ± 8.7		66.2 ± 7.02		0.356
SBP (mmHg)	T1	127.3 ± 7.8	0.334	125.2 ± 6.6	0.914	0.512
	T2	123.8 ± 10.3		126.5 ± 7.3		0.588
DBP (mmHg)	T1	83.2 ± 7.2	0.432	81.1 ± 6.4	0.856	0.634
	T2	81.2 ± 4.1		82.7 ± 4.5		0.912

Intervention (IG) vs. Control (CG): Mann–Whitney test; Assessment Time 1 (T1) vs. Assessment Time 2 (T2): paired Wilcoxon test; * statistically significant (*p* < 0.05); LVEF—left ventricular ejection fraction; GLS—global longitudinal strain (for ease of interpretation, a measure of GLS with negative values here uses positive values); LAVI—left atrial volume index; RVEF—right ventricular ejection fraction; TAPSE—tricuspid annular plane systolic excursion; HR—heart ratio; SBP—systolic blood pressure; DBP—diastolic blood pressure.

**Table 3 jcm-09-01379-t003:** Results of physical capacity tests in the study groups.

Parameter	Time	Intervention Group (IG)		Control Group (CG)		*P_IG_* _*vs. UG*_
		Mean ± SD	*P_T1vsT2_*	Mean ± SD	*P_T1vsT2_*	
6MWD (m)	T1	448.7 ± 50.06	0.312	441.6 ± 24.88	0.042 *	0.314
	T2	449.6 ± 55.33		416 ± 31.68		0.089
MET Unit	T1	3.14 ± 0.24	0.077	3.11 ± 0.12	0.081	0.211
	T2	3.19 ± 0.26		2.93 ± 0.15		0.041 *
Borg scale (point)	T1	1.62 ± 0.72	0.773	1.82 ± 0.6	0.149	0.398
	T2	1.69 ± 0.7		2.09 ± 0.3		0.069

Intervention (IG) vs. Control (CG): Mann–Whitney test; Assessment Time 1 (T1) vs. Assessment Time 2 (T2): paired Wilcoxon test; * statistically significant (*p* < 0.05); 6MWD - 6-Minute Walk Distance; MET—metabolic equivalent of task.

**Table 4 jcm-09-01379-t004:** Comparison of the blood parameters between the study groups.

Parameter	Time	Intervention Group (IG)		Control Group (CG)		*P_IG vs. UG_*
		Mean ± SD	*P_T1vsT2_*	Mean ± SD	*P_T1vsT2_*	
hsCRP (mg/L)	T1	2.2 ± 2.24	0.832	3.1 ± 2.4	0.788	0.565
	T2	3.5 ± 2.7		3.2 ± 3.61		0.988
MYO (pg/mL)	T1	26.04 ± 15.24	0.877	24.78 ± 5.89	0.412	0.517
	T2	28.52 ± 18.67		22.54 ± 4.3		0.219
IL-6 (pg/mL)	T1	4.45 ± 11.81	0.069	3.69 ± 12.92	0.241	0.133
	T2	6.29 ± 12.28		2.61 ± 11.5		0.091
CK (U/L)	T1	29.36 ± 32.34	0.092	32.81 ± 15.63	0.841	0.421
	T2	43 ± 35.96		32.55 ± 17.2		0.138
CK-MB (U/L)	T1	4.61 ± 2.84	0.532	4.47 ± 5.42	0.57	0.71
	T2	5.54 ± 4.13		3.75 ± 6.17		0.317
AST (U/L)	T1	23.91 ± 13.18	0.7	26.02 ± 7.52	0.629	0.144
	T2	27.16 ± 9.41		24.76 ± 8.07		0.691
ALT (U/L)	T1	10.22 ± 5.8	0.997	11.28 ± 5.42	0.998	0.489
	T1	10.21 ± 5.98		11.63 ± 5.62		0.374

Intervention (IG) vs. Control (CG): Mann-Whitney test; Assessment Time 1 (T1) vs. Assessment Time 2 (T2): paired Wilcoxon test; hsCRP—high-sensitivity C-reactive protein; MYO—myoglobin; IL-6—interleukin-6; CK—creatine kinase; CK-MB—creatine kinase myocardial band; AST—aspartate aminotransferase; ALT—alanine aminotransferase.

**Table 5 jcm-09-01379-t005:** Correlation between the blood parameters and BMI, physical capacity, and heart function measurements.

Assessment Time	Group	Parameters					
			BMI	6MWD	Borg Scale	LVEF	MET Units
**T1**	IG	CK-MB	r = 0.707 *	r = −0.694 *	r = 0.536 *	r = −0.568 *	r = −0.694 *
**T2**			r = 0.518 *	r = −0.286	r = 0.286	r = −0.279	r = −0.276
**T1**	CG	CK-MB	-	r = −0.436	-	-	r = −0.436
**T2**			-	r = −0.764 *	-	-	r = −0.764 *
**T1**		IL-6	-	r = −0.764 *	-	-	r = −0.764 *
**T2**			-	r = −0.545	-	-	r = −0.545

r - Spearman’s correlation coefficient; * statistically significant (*p* < 0.05).

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
