# Peer review of "The Preventive Role of Regular Physical Training in Ventricular Remodeling, Serum Cardiac Markers, and Exercise Performance Changes in Breast Cancer in Women Undergoing Trastuzumab Therapy—An REH-HER Study"

_jcm, 2020, doi:10.3390/jcm9051379_

Round 1

Reviewer 1 Report

  1. This phrase". In general opinion, physical exercise training during cancer treatment is very often difficult to apply and clinical elements restricting physical performance such as cardiac dysfunction, fatigue, and other comorbidities should be thoroughly studied" Which general opinion? There are many publications that defend to perform exercise during the treatment as well. 
  2. Can you explain the scheme of chemotherapy received by your patients?
  3. The spelling of English needs revision: from 223 lines to 234. 
  4. Table 1 need to be more " simple",  intuitive and correct the layout,  please check other publications. Please check all the tables.
  5. The discussion needs deep revision  please ( more structure and a link between the paragraphs :
    1. Ex1:  Our exercise program "was" based on the American College of Sports Medicine recommendations [15,22]. line 293
    2. Ex 2:To conduct the evaluation, we used transthoracic Doppler
      echocardiography according to recommendations [25], which was performed only by one and the same physician (cardiologist) to avoid measuring errors. \ The ASE and the European Association of Cardiovascular Imaging define cardiotoxicity as a significant decrease in LVEF >10% or to a value below 53% [10,11]. 
  6. Reference 9, not has DOI

Reviewer 2 Report

In this study Hojan et al., describe the significance of preventive role of regular physical training in ventricular remodeling, serum cardiac markers and exercise performance changes in breast cancer women undergoing trastuzumab therapy. Specifically, they try to link the relationship between supervised and moderate-intensity exercise training would prevent heart failure and its consequences induced by trastuzumab therapy. They observed that moderate-intensity exercise training prevents a decrease in LVEF and physical capacity during trastuzumab therapy in HER2+ BC.

In summary, this is an interesting study with great potential. Overall, this study conducted precisely and appears to be planned and the results support the conclusion. The experiments are strikingly presented, and the results appears solid and their analyses are reasonable. Simultaneously, I have some suggestions need to be addressed by authors.

Comments:

  1. In the introduction section need the very recent literature about incidence and mortality rate of breast cancer cases globally.
  2. In the introduction section need the some literature about different strategies used to treatment breast cancer in preclinical models by citing these papers in addition with others. (Bitter melon extract inhibits breast cancer growth in preclinical model by inducing autophagic cell death. Oncotarget. 2017 Aug 03. doi: 10.18632/oncotarget.19887, Strategy to enhance the efficacy of doxorubicin in breast and hepatocellular carcinoma cells by methyl- β-cyclodextrin: Activation of p53 and involvement of Fas receptor ligand. Scientific Reports 2015 7;5:11853. doi: 10.1038/srep11853, Anti-miR-203 suppresses breast cancer growth and stemness by targeting SOCS3. Oncotarget. 2016 Aug 10. doi: 10.18632/oncotarget.11193).
  3. It would be very nice, if authors could make the bar graph in place of table 4.
  4. Authors have not done extensive revision of manuscript in terms of the language and sentence forming.  It does not look good when you read a manuscript in this kind of language. Please work on the manuscript writing part. There are so many places where unnecessary words have been used.
  5. Please try to correct typological errors throughout the manuscript. I found many typological errors at various places.
  6. References should be written according to the journal guideline

Author Response

We would like to thank you for your suggestions. We have made our point-to-point replies below. Please take a look at our comments below.

Comments:

  1. In the introduction section need the very recent literature about incidence and mortality rate of breast cancer cases globally.

Thank you for this comment. We have added this data in the introduction.Breast cancer is the most common cancer affecting women, causing 24.6% of female cancers and resulting in 15% of all female cancer deaths worldwide.”

  1. In the introduction section need the some literature about different strategies used to treatment breast cancer in preclinical models by citing these papers in addition with others. (Bitter melon extract inhibits breast cancer growth in preclinical model by inducing autophagic cell death. Oncotarget. 2017 Aug 03. doi: 10.18632/oncotarget.19887, Strategy to enhance the efficacy of doxorubicin in breast and hepatocellular carcinoma cells by methyl- β-cyclodextrin: Activation of p53 and involvement of Fas receptor ligand. Scientific Reports 2015 7;5:11853. doi: 10.1038/srep11853, Anti-miR-203 suppresses breast cancer growth and stemness by targeting SOCS3. Oncotarget. 2016 Aug 10. doi: 10.18632/oncotarget.11193).

Following the Reviewer’s suggestion, we have added data in the introduction about current preclinical studies. Please see lines:42-43.

  1. It would be very nice, if authors could make the bar graph in place of table 4.

Thank you for this suggestion. After careful discussions, the authors of the manuscript decided to simplify the layout of the tables, providing a consistent presentation of the results.

  1. Authors have not done extensive revision of manuscript in terms of the language and sentence forming.  It does not look good when you read a manuscript in this kind of language. Please work on the manuscript writing part. There are so many places where unnecessary words have been used.

Thank for this suggestion, we have made an extensive revision of the manuscript with the help of a professional linguist using MPDI English Service.

  1. Please try to correct typological errors throughout the manuscript. I found many typological errors at various places.

Thank you for this comment. We have revised the manuscript and corrected the typological errors.

  1. References should be written according to the journal guideline

Thank you for this comment. We have rewritten the references in accordance with the journal guidelines.

Round 2

Reviewer 1 Report

Has been improved since first manuscript. 

The table 1 has to be clearer

Author Response

We would like to thank the Reviewer for his  comments, which helped us to rewrite and improve the manuscript. 

We have tried our best to present the data of the study participants in Table 1. We hope that the current changes are sufficient. Thank you for this comment.

Complying with your suggestions, we have revised themanuscript in English language and style.

Authors